# Loss of H3K27me3 imprinting in the Sfmbt2 miRNA cluster causes enlargement of cloned mouse placentas

Kimiko Inoue [1,2✉], Narumi Ogonuki[1], Satoshi Kamimura[1,7], Hiroki Inoue[1,8], Shogo Matoba [1,3], Michiko Hirose[1], Arata Honda[1,4], Kento Miura[1], Masashi Hada[1], Ayumi Hasegawa[1], Naomi Watanabe [1,2], Yukiko Dodo[1], Keiji Mochida[1] & Atsuo Ogura[1,2,5,6✉]

Somatic cell nuclear transfer (SCNT) in mammals is an inefficient process that is frequently associated with abnormal phenotypes, especially in placentas. Recent studies demonstrated that mouse SCNT placentas completely lack histone methylation (H3K27me3)-dependent imprinting, but how it affects placental development remains unclear. Here, we provide evidence that the loss of H3K27me3 imprinting is responsible for abnormal placental enlargement and low birth rates following SCNT, through upregulation of imprinted miRNAs. When we restore the normal paternal expression of H3K27me3-dependent imprinted genes (*Sfmbt2*, *Gab1*, and *Slc38a4*) in SCNT placentas by maternal knockout, the placentas remain enlarged. Intriguingly, correcting the expression of clustered miRNAs within the *Sfmbt2* gene ameliorates the placental phenotype. Importantly, their target genes, which are confirmed to cause SCNT-like placental histology, recover their expression level. The birth rates increase about twofold. Thus, we identify loss of H3K27me3 imprinting as an epigenetic error that compromises embryo development following SCNT.

[1] Bioresource Engineering Division, Bioresource Research Center, RIKEN, Tsukuba, Ibaraki 305-0074, Japan. [2] Graduate School of Life and Environmental Sciences, University of Tsukuba, Tsukuba, Ibaraki 305-8572, Japan. [3] Cooperative Division of Veterinary Sciences, Tokyo University of Agriculture and Technology, Fuchu, Tokyo 183-8509, Japan. [4] Institute of Laboratory Animals Graduate School of Medicine, Kyoto University, Kyoto 606-8501, Japan. [5] RIKEN Cluster for Pioneering Research, Wako, Saitama 351-0198, Japan. [6] The Center for Disease Biology and Integrative Medicine, Faculty of Medicine, University of Tokyo, Tokyo 113-0033, Japan. [7] Present address: Department of Basic Medical Sciences for Radiation Damages, National Institute of Radiological Sciences, National Institutes for Quantum and Radiological Science and Technology, Chiba 263-8555, Japan. [8] Present address: Department of Gene Function and Phenomics, National Institute of Genetics, Mishima, Shizuoka 411-8540, Japan. ✉email: inoue@rtc.riken.jp; ogura@rtc.riken.go.jp

The placenta is an organ that provides the interface for gas and nutrition exchange between the mother and fetus. The placenta also serves as an immunological barrier to protect the fetus from immunological attack by the mother. Therefore, the functions and structural integrity of the placenta are maintained by fine-tuned regulatory mechanisms on both the mother and fetal sides at the cellular, endocrine, and immunological levels. If this balance is disturbed, the development of the fetus, or even the health status of the mother, may be compromised. For example, in humans, dysfunctional placental development may lead to preeclampsia and hemolysis, elevated liver enzymes, and low platelet count (HELLP) syndrome[1], which is often associated with severe maternal hypertension.

Somatic cell nuclear transfer (SCNT) is the only reproductive technique that can generate animals from oocytes and single cells. It is known that SCNT is frequently associated with abnormal phenotypes, especially those related to placental development[2]. Even in cattle and sheep, the most successfully cloned species, SCNT remains associated with a high incidence of abnormal placental development events, such as reduced vascularization and fewer but enlarged placentomes[3]. Cloned mice also exhibit a definite type of placental abnormality, i.e., extraordinary enlargement (placental hyperplasia) with an expanded spongiotrophoblast (ST) layer in all cases, except in those with a specific genetic background (the 129 strain), for unknown reasons[4–6]. Intriguingly, none of the recent technical breakthroughs, such as histone deacetylase treatment, correction of *Xist* expression, removal of repressive histone H3K9me3, or their combination, can ameliorate SCNT-specific placental abnormalities, even though the birth rates are currently as high as 18.7% (refs. [7–9]). Transcriptome and DNA methylation analyses identified the dysregulated expression of specific genes (*Car2* and *Ncam1*) and the hypermethylation of the *Sall3* locus, but they were considered to be downstream events relative to placental hyperplasia[10,11]. Thus, the precise etiology of the placental enlargement in cloned mice remains unclear.

Genomic imprinting in mammals is an epigenetic process, where the two parental alleles of a gene are differentially expressed. The parent of origin-specific monoallelic expression of imprinted genes is mediated by an imprinting control region (ICR), which possesses parent-specific differential epigenetic marks, mostly DNA methylation[12]. This DNA methylation pattern is established during gametogenesis, and it is thought to be maintained to support embryonic and placental development[13]. Therefore, the loss of imprinting (LOI) induces no or biallelic expression of the imprinted genes, which may cause developmental abnormalities. In SCNT placentas, three placenta-specific imprinted genes (*Sfmbt2*, *Gab1*, and *Slc38a4*) were biallelically expressed due to LOI (ref. [14]). Recently, it was demonstrated that these genes are primarily imprinted by maternal allele-specific trimethylation at lysine 27 in histone H3 (H3K27me3)[15], and that this H3K27me3 mark is lost completely in the extraembryonic linage of SCNT embryos[16]. Previous gene knockout (KO) experiments showed that at least two of these imprinted genes (*Sfmbt2* and *Gab1*) are essential for normal placental development, and thus their biallelic expression mediated by LOI may cause SCNT-associated placental hyperplasia[17,18].

MicroRNAs (miRNAs) are short noncoding single-stranded RNAs that suppress gene expression by inhibiting translation or reducing the stability of target mRNAs. More than 1000 miRNAs are found in mammals and they play pivotal roles in a wide variety of biological processes, such as cell proliferation, apoptosis, tumorigenesis, and metabolism. Many miRNAs are known to be critically involved in the placental development via these processes[19], and some are under the control of imprinted genes[20]. For example, the antisense of the *Rtl1* (*Peg11*) imprinted gene

encodes seven clustered miRNA genes, and their depletion induces placental hyperplasia and neonatal death[21,22]. The paternally imprinted gene *H19* contains *miR-675*, which negatively regulates the placental development[23]. Interestingly, these miRNAs and/or their host genes are aberrantly regulated in SCNT embryos or placentas[24,25].

In this study, we aim to identify the primary cause of SCNT-associated placental hyperplasia in mice by focusing on the possible involvement of LOI for H3K27me3, and the dysregulation of miRNAs expressed in the placenta. By combining transcriptome analysis with gene KO experiments, we demonstrate that the upregulation of H3K27me3-dependent imprinted miRNAs, rather than protein-coding imprinted genes, is responsible for placental hyperplasia in mouse SCNT. This discovery advances our understanding of how miRNAs regulate placental development and consequently embryo development via the postulated combined effects of a large set of their target genes.

## Results

**H3K27me3 imprinted genes are not cause of enlarged placentas.** First, we attempted to identify the H3K27me3-dependent imprinted genes responsible for SCNT-associated placental hyperplasia. In normal placentas, *Sfmbt2*, *Gab1*, and *Slc38a4* are paternally expressed, whereas they are biallelically expressed in SCNT placentas due to LOI (refs. [14,16]). Therefore, we restored their normal paternal expression in SCNT placentas using cumulus cells from donor mice carrying the maternal KO allele for each of these separate imprinted genes, and then examined the resulting placental size. The wild type SCNT placentas weighed $0.328 \pm 0.02$ g (mean ± standard error of the mean (SEM)), which was significantly greater than that of the in vitro fertilization (IVF)-derived placentas ($0.107 \pm 0.004$ g, $P < 0.0001$, Kruskal–Wallis test; Fig. 1a). The SCNT placentas that carried the maternal KO for *Sfmbt2*, *Gab1*, and *Slc38a4* weighed $0.280 \pm 0.025$, $0.322 \pm 0.036$, and $0.280 \pm 0.010$ g, respectively, which were significantly different from the weights of the IVF-derived placentas (Fig. 1a). Placentas from these maternal KO maintained the SCNT-specific placental histology (e.g., *Sfmbt2* maternal KO in Fig. 1b). Our quantitative RT-PCR (qRT-PCR) analysis confirmed that the three genes that were specifically upregulated by wild type SCNT were corrected by maternal SCNT KO (Supplementary Fig. 1). Thus, the biallelic expression of any of the H3K27me3-dependent imprinted genes that we examined was not a primary cause of placental hyperplasia in SCNT.

**Dysregulation of imprinted miRNA clusters in SCNT placentas.** Next, we examined whether a subset of miRNAs were differentially expressed in the enlarged SCNT placentas. We analyzed the miRNA expression profiles in the SCNT placentas using Agilent SurePrint G3 mouse miRNA microarrays. To avoid the possible transcriptome bias caused by the altered histology in the SCNT placentas, we aimed to identify the appropriate gestational day to facilitate the comparative analysis of the IVF-derived and SCNT-derived placentas. It is known that early SCNT placentas (E6.5–9.5) consistently exhibit poor development because of the slow proliferation of trophoblastic cells[26,27], whereas they proliferate rapidly in the later gestational stages (E13.5–19.5). We found that SCNT placentas at E11.5 were identical to age-matched IVF-derived placentas in terms of their size and histology (Fig. 2a, b). Therefore, we identified miRNAs with aberrant expression in cloned placentas by comparing miRNA transcriptome data obtained from E11.5 placentas generated by SCNT, using wild type and *Xist*-KO cumulus cells ($N = 3$ and 2, respectively) or Sertoli cells ($N = 3$ and 2, respectively), with those from age-matched IVF-derived placentas ($N = 4$).

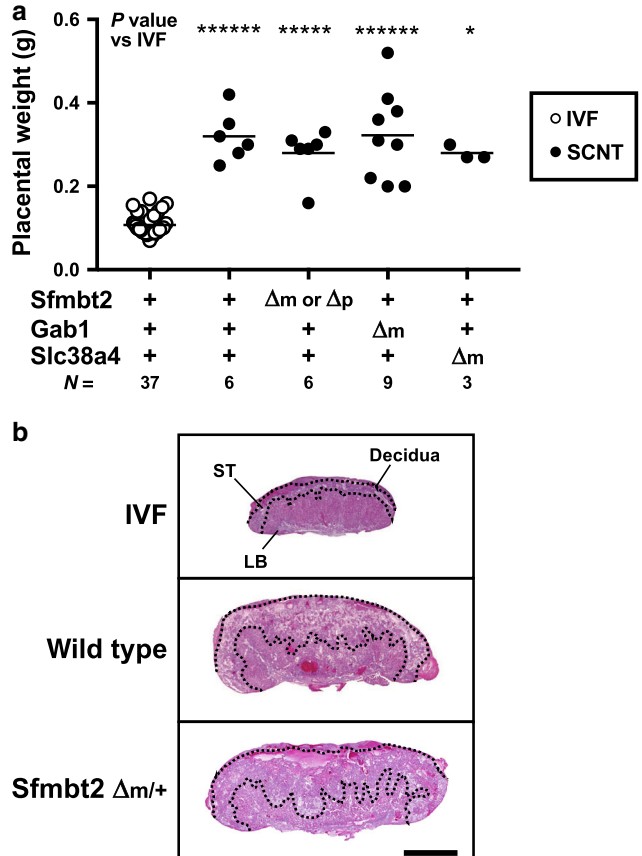

**Fig. 1 Placental weights and histology of IVF and SCNT from wild type or KO mice. a** Weights of term placentas derived from IVF or SCNT. The horizontal lines indicate the mean value. +, wild type; Δm, maternal KO; Δm/p, maternal or paternal KO; *$P < 0.05$, *****$P < 0.0005$, ******$P < 0.0001$ (Kruskal–Wallis test). N represents the number of biological replicates. *Gab1* KO included six placentas cloned from Sertoli cells. Source data are provided as a Source data file. **b** Hematoxylin and eosin-stained tissue sections of E19.5 placentas from IVF and SCNT (wild type and *Sfmbt2* maternal KO placentas). ST spongiotrophoblast layer, LB labyrinthine layer. Scale bar, 2 mm.

Among the 1079 miRNAs analyzed, 98 and 32 miRNAs were differentially expressed between IVF and cumulus- or Sertoli-cloned placentas, respectively (Supplementary Data 1), where 21 genes were common to both groups (Fig. 2c). In particular, 15 out of 21 genes were located on chromosome 2 (2qA1) or chromosome 12 (12qF1) (Fig. 2d). These two regions correspond to large miRNA clusters in mice, which are located within imprinting domains. Interestingly, the miRNA cluster on 2qA1 was located in the intron for *Sfmbt2*, which is a H3K27me3-dependent imprinted gene[28,29]. This is the largest miRNA cluster in mice and it contains 72 miRNA precursor sequences. The other cluster on 12qF1 was located with the *Mirg* gene, a maternally expressed gene within the *Dlk-Dio3* domain[30]. In the following, these clusters are referred to as the *Sfmbt2* and *Mirg* miRNAs, respectively. The altered expression levels of these miRNAs might reflect the status of their host gene (*Sfmbt2*) and the imprinting center (intergenic differentially methylated region), respectively, because both are known to exhibit LOI in SCNT placentas[14] and SCNT-derived trophoblast stem cells (TSCs)[31]. Scatterplots showing the variations in the expression of all the miRNA genes between IVF and wild type SCNT placentas clearly distinguished the *Sfmbt2* miRNAs and *Mirg* miRNAs from other genes based on their upregulation and downregulation, respectively (Fig. 3a, b). These trends remained consistent even when we used

*Xist*-KO donor cells, which significantly improved the cloning efficiency by correcting the ectopic expression of *Xist* in SCNT embryos (Supplementary Fig. 2a, b). The expression levels of representative miRNA genes are illustrated in Fig. 3c and Supplementary Fig. 2c. In SCNT placentas, the *Sfmbt2* miRNAs were upregulated regardless of the donor cell type, whereas the *Mirg* miRNAs often exhibited a donor cell-specific pattern, with lower expression in Sertoli cell-derived placentas compared with cumulus cell-derived placentas probably because of some regulatory mechanisms specific to the *Dlk1–Dio3* imprinted domain (Fig. 3c, Supplementary Fig. 2c). We also confirmed the dysregulation of these miRNAs in SCNT-derived TSCs, although some gene-specific variations were observed (Supplementary Fig. 3).

Previously, we identified the essential roles of *Sfmbt2* miRNAs in normal placental development by deleting the entire *Sfmbt2* miRNA cluster (~53 kb)[32]. Importantly, the region affected most by miRNA depletion was the ST layer, which agreed exactly with that in SCNT placentas. By contrast, the *Mirg* miRNAs seemed to have little or no impact on the placental development because maternal KO only yielded postnatal phenotypes, such as energy homeostasis disorder and loss of sociability[33,34]. Therefore, we focused on *Sfmbt2* miRNAs in the subsequent analyses and assessed their possible involvement in SCNT-associated placental hyperplasia.

**Sfmbt2 miRNAs are expressed predominantly in the ST layer.** To determine the localization of the *Sfmbt2* miRNAs in the placenta, we performed in situ hybridization in E11.5 IVF-derived placentas (Fig. 4a), using specific probes for the highly expressed *miR669f-3p* gene (Fig. 3c). Positive staining of nuclei was identified in secondary trophoblast giant cells (Fig. 4b, c). We also found positive nuclear staining in trophoblasts from the ST layer, and immature trophoblasts from the labyrinthine (LB) layer (Fig. 4d, e). Their cytoplasm was faintly stained. The ST layer was the region affected most severely by the deletion of the *Sfmbt2* miRNAs[32].

**Sfmbt2 miRNAs correction ameliorates placental hyperplasia.** To assess the possible involvement of the upregulated *Sfmbt2* miRNAs in SCNT-specific placental hyperplasia, we produced SCNT placentas with corrected expression of the *Sfmbt2* miRNAs using a mouse line that lacked the entire *Sfmbt2* miRNA cluster (henceforth referred to as miRNA KO)[32]. The SCNT placentas with maternal miRNA KO weighed $0.20 \pm 0.01$ g and they did not differ significantly from the weights of the IVF-derived placentas ($P = 0.585$, Kruskal–Wallis test; Fig. 5a). This suggests that upregulation of the *Sfmbt2* miRNAs was at least partly responsible for the enlargement of the cloned placentas. The average birth weight of miRNA KO pups was not different from that of IVF-derived or wild type cloned pups ($1.54 \pm 0.04$ g, $1.73 \pm 0.10$ g, and $1.63 \pm 0.08$ g in IVF-derived, wild type, and miRNA KO pups, respectively). These miRNA KO cloned pups were normal in appearance and grew into normal fertile adults. Subsequently, we tested whether there was any combined effect of the *Sfmbt2* miRNAs and *Gab1* because *Gab1* is known to affect the placental development via gene deletion[18]. Intriguingly, the placental weight was decreased further under these conditions ($0.17 \pm 0.01$ g, $P = 0.981$ vs IVF, Kruskal–Wallis test, $P = 0.048$ vs miRNA KO, $t$-test, Fig. 5a).

Next, we assessed the histological changes in SCNT placentas with or without gene modifications. Wild type cloned placentas were characterized by expansion of the ST layer with increased glycogen cells, and by distortion of the boundary between the ST and LB layers (Fig. 5b, c)[27]. By contrast, a significant reduction in the ST layer was observed in the SCNT placentas with miRNA KO and in those with miRNA/*Gab1* double KO ($P < 0.0005$ in

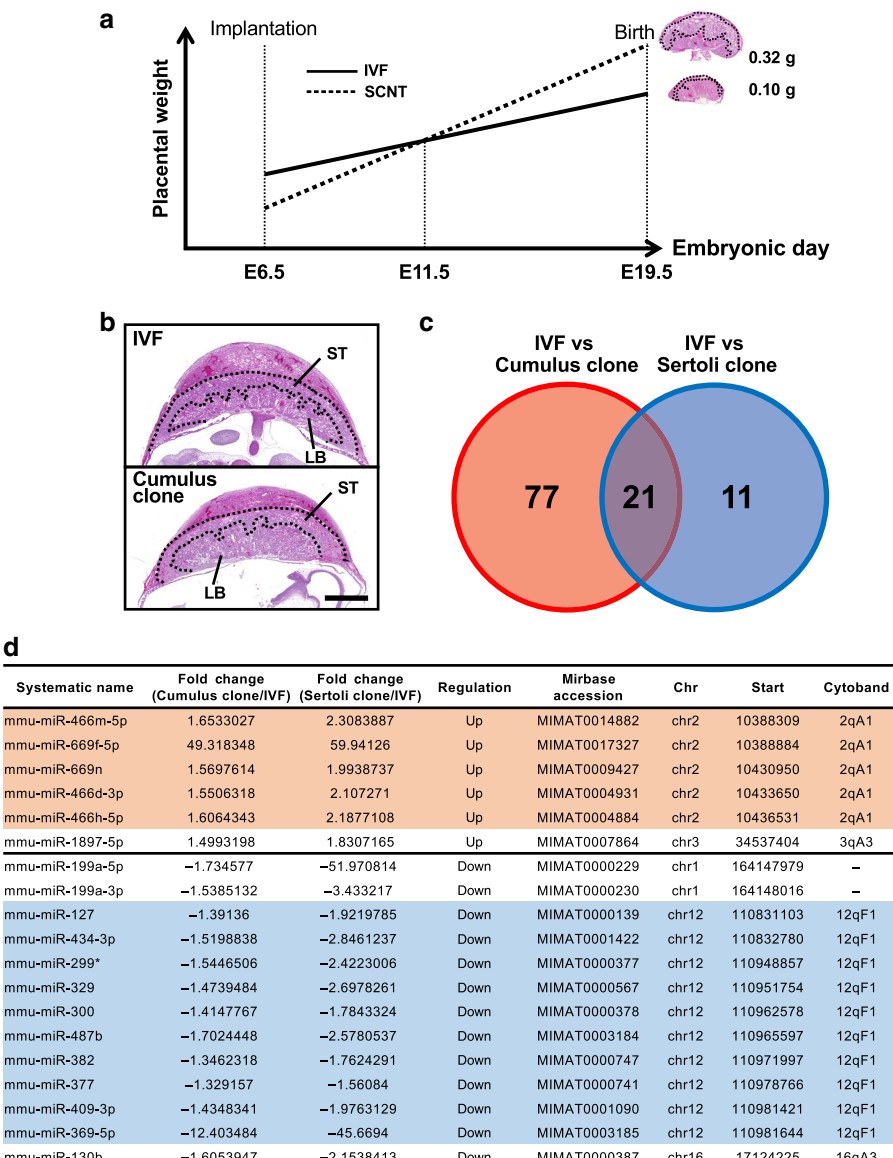

**Fig. 2 Expression analysis of miRNAs in IVF- and SCNT-derived placentas at E11.5. a** Changes in the size of IVF- and SCNT-derived placentas during gestation. SCNT placentas were smaller than IVF placentas in the early stage but larger in the later stage. IVF and SCNT placentas at E11.5 were identical in size and structure, and they were considered to be appropriate for comparative transcriptome analysis, as shown in **b**. **b** Histology of IVF and SCNT placentas at E11.5. ST spongiotrophoblast layer, LB labyrinthine layer. Scale bar, 1 mm. **c** Venn diagram showing the numbers of miRNAs with differential expression levels in E11.5 SCNT placentas derived from cumulus and Sertoli cells. Twenty-one differentially expressed miRNAs were common to the two types of SCNT placentas. **d** List of the 21 common differentially expressed miRNAs. Rows in orange and blue indicate genes located on chromosome 2 (2qA1) and chromosome 12 (12qF1), respectively. Annotations were determined according to the information from Agilent SurePrint G3 mouse miRNA microarray miRBase rel.17 version. Mmu-miR-199a-5p and -3p are annotated on chromosome 9 in the current database. Also see Supplementary Data 1.

wild type, $P < 0.05$ in miRNA KO, and $P = 0.1641$ in miRNA/ *Gab1* double KO (each vs IVF), two-way ANOVA, Fig. 5b, c). Therefore, maternal KO prevented the expansion of the ST layer, which occurs specifically in SCNT placentas. However, the distortion of the boundary between the ST and LB layers persisted in the KO placentas, thereby indicating that this abnormality was a consequence of other unknown mechanisms (Fig. 5b, c).

**Gene expression pattern of miRNA KO SCNT placentas.** To understand the transcriptomic changes caused by SCNT in placentas, we conducted RNA-seq analysis using E11.5 and E19.5

(term) placentas derived from IVF and SCNT with or without gene KO (total of eight groups and 22 samples). Principal component analysis mapped the E11.5 and E19.5 placentas separately along Component 1 (Fig. 6a). As expected, the E11.5 placentas were grouped together more closely, thereby indicating that they had less transcriptomic variation than the E19.5 placentas. In the E11.5 group, the IVF placentas and wild type placentas were clearly separated along Component 1, and it was important that the SCNT placentas with miRNA KO or miRNA/*Gab1* double KO were positioned between them, which indicates that deletion of the miRNAs yielded gene expression patterns closer to those of the IVF placentas. In the E19.5 group, the IVF and SCNT

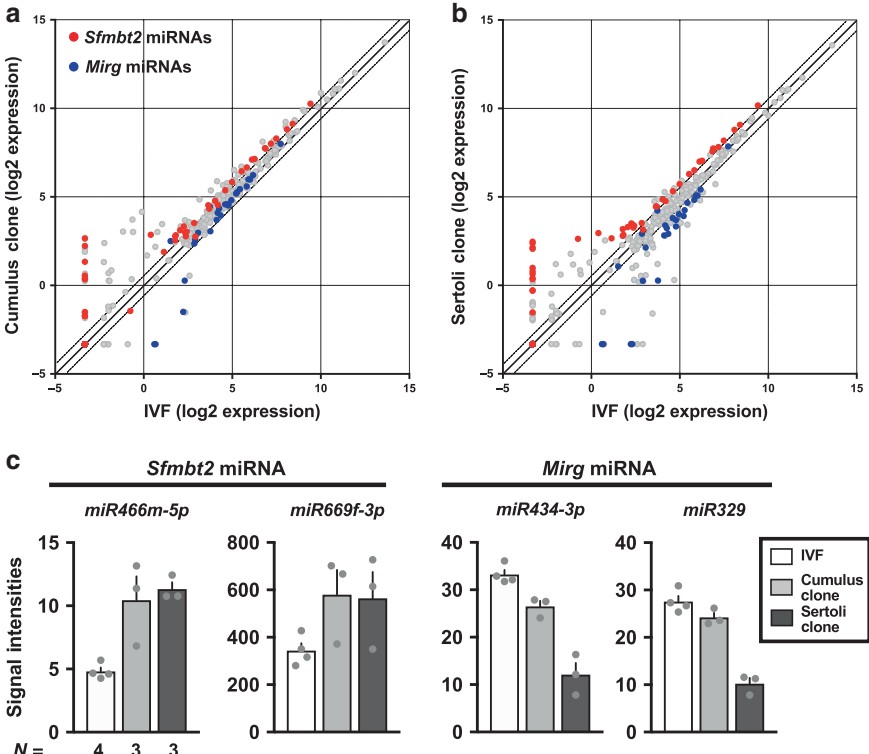

**Fig. 3 Dysregulation of two clustered miRNAs in SCNT placentas. a, b** Scatterplot analysis of all miRNAs based on their expression levels in cumulus- **a** and Sertoli- **b** derived SCNT placentas compared with those detected in IVF placentas. Red and blue dots represent miRNAs from the *Sfmbt2* and *Mirg* clusters, respectively. Dotted lines represent >1.3-fold changes. **c** Expression levels of *Sfmbt2* and *Mirg* miRNAs. The *Sfmbt2* miRNAs were highly expressed in SCNT placentas, regardless of the donor cell type. The *Mirg* miRNAs had lower expression levels in SCNT placentas and they exhibited donor cell type dependency. Also see Supplementary Figs. 2 and 3. *N* represents the number of biological replicates. The error bars represent the SEM. The Source data of Fig. 2c are provided as a Source data file.

placentas were also separated along Component 1. Three groups of SCNT placentas were not separated by Component 1, whereas the wild type and two types of KO SCNT placentas were separated by Component 2.

To obtain insights into the gene expression dynamics that accompanied the changes in *Sfmbt2* miRNA expression, we focused on the differentially expressed genes (DEGs) between wild type SCNT and IVF placentas (fold change > 1.8), which comprised 1257 genes at E11.5, and 1319 genes at E19.5 (Fig. 6b, Supplementary Data 2–5). In particular, after miRNA KO, the numbers of genes decreased to 395 and 521 at E11.5 and E19.5, respectively (Fig. 6b). After miRNA/*Gab1* double KO, the numbers of genes also decreased to 495 and 522, respectively (Fig. 6b). The top ten Gene Ontology (GO) terms and the *P*-values for DEGs between the wild type SCNT and IVF placentas are indicated in Supplementary Fig. 4. The genes upregulated at E11.5 appeared to be correlated with ion transport, whereas the downregulated genes included terms related to immune system process and female pregnancy. The enrichment of GO terms decreased in the miRNA or double KO SCNT placentas, with only a few exceptions (e.g., the GO term "female pregnancy" for genes that were downregulated at E11.5).

As expected, the expression levels of the *Sfmbt2* miRNAs (specifically pri-miRNAs, e.g., *Mir467b*, *Mir297b*, and *Mir669a-3*) were normalized in the KO and double KO SCNT placentas (Fig. 6c, Supplementary Fig. 5). The *Sfmbt2* and *Gab1* genes were also included in the group of DEGs upregulated at E11.5 and E19.5, which was predicted from their LOI (Fig. 6c). As expected, *Sfmbt2* remained highly expressed in the two types of KO SCNT placentas, and *Gab1* was normalized in the *Sfmbt2* miRNA/*Gab1*

double KO SCNT placentas after maternal allele deletion. Other placenta-specific imprinted genes, *Smoc1* and *Phf17* (ref. [15]), were also upregulated in E11.5 SCNT placentas (Supplementary Fig. 5), whereas the expression levels of *Smoc1* in E19.5, and of *Phf17* in E11.5 placentas seemed to be corrected by miRNA KO and miRNA/*Gab1* KO placentas. Although the reasons for their corrected expression patterns are unknown, a normalized placental histology (especially at E19.5) or corrected physiological conditions may have been at play. *Slc38a4* was not upregulated in SCNT placentas, which may have been caused by the great level of variation observed in the IVF placentas. Our qRT-PCR analysis revealed that *Slc38a4* was significantly upregulated in SCNT placentas at E19.5 (Supplementary Fig. 1). We also reported previously that this gene was upregulated in E13.5 SCNT placentas[14]. We also observed changes in the expression levels of canonical imprinted genes in E11.5 placentas. Two genes (*Dio3* and *Phlda2*) were upregulated and eight genes (*Rasgrf1*, *Gatm*, *Dcn*, *Slc22a2*, *Tnfrsf23*, *Ampd3*, *Zim1*, and *Tnfrsf22*) were downregulated (Fig. 7a). Interestingly, the latter downregulated genes, but not the former upregulated genes, tended to exhibit improved expressions by *Sfmbt2* miRNA KO and *Sfmbt2* miRNA/*Gab1* KO (Fig. 7a).

We also noted a bias in the chromosomal locations of DEGs, i.e., 13qA3.1 ($P < 0.005$, Fisher's exact test) for the upregulated genes and 7qA2 ($P < 1.0 \times 10^{-4}$) for the downregulated genes at E11.5. These regions include clusters of prolactin (*Prl*) family genes and specific glycoprotein (*Psg*) family genes, respectively (Supplementary Data 6). These gene families play important roles in the development of the placenta and the maintenance of pregnancy[35–37]. Our expression analysis results suggested that

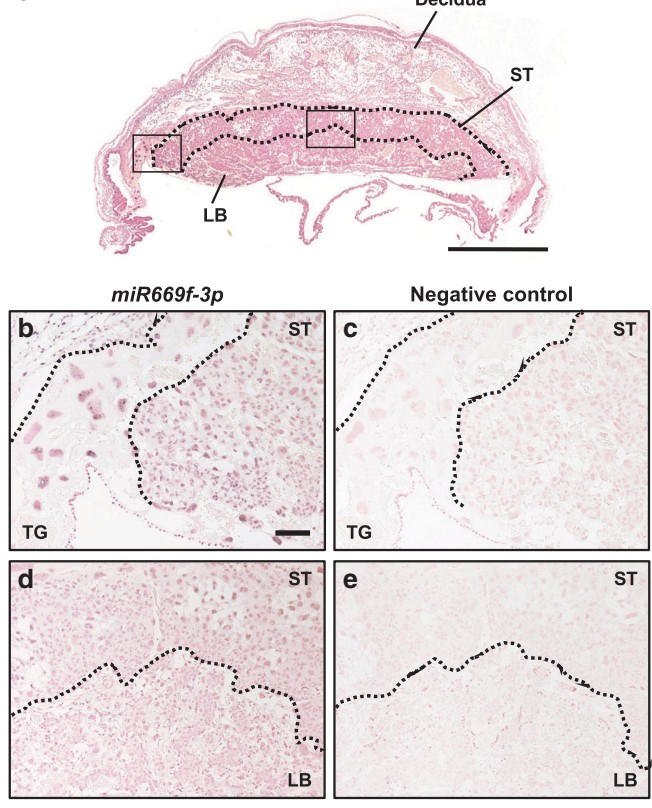

**Fig. 4 Localization of *miR669f-3p*, *Sfmbt2* miRNA, in E11.5 placenta. a** In situ hybridization image of a normal (IVF-derived) placenta at E11.5. The areas corresponding to the magnified images in **b** and **c** are indicated by black squares. ST spongiotrophoblast layer, LB labyrinthine layer. Scale bar, 1 mm. **b**–**e** In situ hybridization using probes for *miR669f-3p* and negative-control probes. The nuclei in the cell layer of secondary trophoblast giant cells (TG) were clearly stained for *miR669f-3p* **b**. The nuclei of trophoblasts in the ST layer and immature trophoblasts in the LB layer **d** were also clearly stained. Their cytoplasm was faintly stained. Scale bar, 100 μm.

both gene families were affected by SCNT independently of the *Sfmbt2* miRNAs because their expression levels were not changed by miRNA KO.

**Target genes of miRNAs are upregulated in KO SCNT placentas.** According to the predicted sequence matches, thousands of genes can be considered targets of the *Sfmbt2* miRNAs. In particular, 15 genes were downregulated in SCNT placentas (Fig. 7b). Importantly, the expression of all 15 genes increased in the miRNA KO and/or miRNA/*Gab1* double KO SCNT placentas (Fig. 7b). In our previous study of *Sfmbt2* miRNA KO mice, we identified the physiologically important target genes based on their altered expression levels in KO placentas at E8.5 and E11.5 (ref. [32]), which included at least four genes that play roles in tumor growth repression or apoptosis (*Gkn2*, *Rgs4*, *Fndc1*, and *Fst*); therefore, they are likely to regulate the size of the placenta. In the present study, three of these genes (*Gkn2*, *Rgs4*, and *Fst*) were included in the list of downregulated target genes, and they exhibited increased expression levels after miRNA KO and miRNA/*Gab1* double KO (Fig. 7b). We used an established KO mouse line (*Gkn2*) or founder (F0) mice generated with the triple-target CRISPR system (*Rgs4* and *Fst*) to analyze the functions of these three genes in placental development. The triple-target CRISPR system allowed us to investigate the KO phenotypes in the F0 generation due to its high KO efficiency[38]. We

found no apparent abnormalities in the placentas of the *Gkn2* and *Rgs4* KO mouse strains, but the *Fst* KO exhibited expansion of the ST layer with an increased number of glycogen cells, which were reminiscent of SCNT placentas (Fig. 7c). We then generated KOs for the other six target genes listed in Fig. 7b and examined their placental phenotypes. Intriguingly, although the placental size did not appear to change significantly, histological analysis of the placentas from all six KOs detected abnormal phenotypes that resembled those of SCNT placentas, including enlargement of the ST layer, increased numbers of glycogen cells, and/or distortion of the boundary between the ST and LB layers (Supplementary Fig. 6). The levels of expression of the mRNA and/or protein of target genes in triple-target CRISPR KO placentas were significantly decreased ($P < 0.05$, *t*-test, Supplementary Fig. 7a, b). Furthermore, KO of the targeted genes that are known to be essential for neonatal development (*Fst* and *Bmper*) caused death of all fetuses shortly after birth[39,40] (Supplementary Fig. 7c, Supplementary Data 7). Thus, the abnormal histology of the SCNT placentas could be explained by downregulation of these genes, as a consequence of the upregulation of *Sfmbt2* miRNAs.

**Birth rates of clones are improved by miRNA/*Gab1* KO.** Finally, we investigated whether the correction of placental hyperplasia could improve the birth rate by the SCNT-derived animals. The birth rates of the clones from miRNA KO (6.7%) and miRNA/*Gab1* double KO (6.7%) donor cells were more than double those of the clones from wild type donors (3.0%), although the differences were not statistically significant ($P = 0.29$ and 0.18, respectively; chi-squared test; Table 1).

**Discussion**
Two key questions have remained unanswered since the first successful SCNT in mammals: (1) why is SCNT frequently associated with placental abnormalities? and (2) are these placental abnormalities responsible for the poor development of clones? In the present study, we obtained important insights to help answer these questions.

The birth rates of clones have increased because of attempts to improve the epigenetic status of cloned embryos[41,42], but the SCNT-specific placental phenotypes remain unresolved, thereby indicating the presence of unidentified epigenetic alterations that are resistant to these epigenetic treatments in SCNT placentas. Therefore, we assumed that the previously reported LOI of H3K27me3-dependent imprinted genes (such as *Sfmbt2*, *Gab1*, and *Slc38a4*)[14] might be responsible for placental hyperplasia in mouse SCNT. However, in the present study, we showed that none of these genes alone is the primary cause of placental hyperplasia based on maternal KO SCNT experiments. Instead, we found that the upregulation of clustered miRNAs located within an intron of *Sfmbt2* was the major cause of placental hyperplasia in cloned mice. Although we could not confirm their biallelic expression because of technical difficulties, it probably occurred because their host *Sfmbt2* gene was biallelically expressed by LOI (refs. [14,16]). Our idea that the upregulation of *Sfmbt2* miRNAs is the major cause of SCNT-associated placental hyperplasia was also supported by KO experiments with their target genes. We confirmed that KO of at least 7 out of 15 downregulated target genes caused enlargement of the ST layer, increases in the number of glycogen cells, and/or distortion of the boundary between the ST and LB layers, which resembled those found in SCNT placentas. It is highly probable that downregulation of these genes led to cumulative increases in placental size via expansion of the ST layer. Thus, we identified the set of genes that directly cause SCNT-specific placental histological features, which is a question that has long remained unanswered[5].

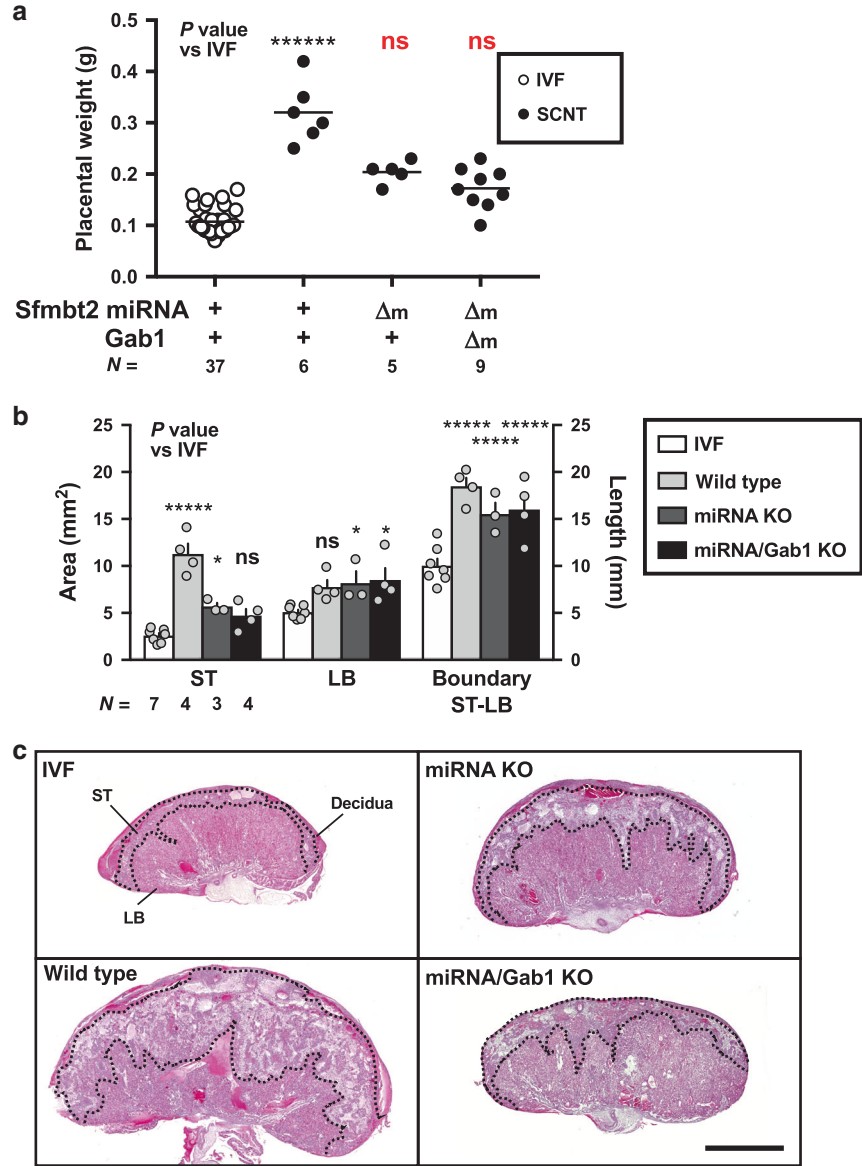

**Fig. 5 Placental weights and histology of SCNT from miRNA KO mice. a** Weights of term placentas derived from IVF or SCNT. The horizontal lines indicate the mean value. +, wild type; Δm, maternal KO; ******$P < 0.0001$, ns, not significantly different (Kruskal–Wallis test). $N$ represents the number of biological replicates. Weights of IVF and wild type SCNT placentas were same as Fig. 1a. **b** Areas of the ST and LB layers, and length of the boundary between ST and LB layers. *$P < 0.05$, *****$P < 0.0005$ (two-way ANOVA). $N$ represents the number of biological replicates. The error bars represent the SEM. **c** Hematoxylin and eosin-stained tissue sections of E19.5 placentas from IVF and SCNT (wild type and two types of KO placentas). ST spongiotrophoblast layer, LB labyrinthine layer. Scale bar, 2 mm. The Source data of **a** and **b** are provided as a Source data file.

Importantly, the birth rates of clones were improved more than twofold by miRNA KO and miRNA/*Gab1* double KO, although the differences were not significant. More replicates might be required to obtain a statistically significant effect. Nonetheless, we identified loss of H3K27me3 imprinting as an epigenetic error that may affect the placental development, and possibly the development of embryos following SCNT. According to our observations, loss of H3K27me3 imprinting affected embryo development to a lesser extent than previously identified epigenetic errors, such as the ectopic expression of *Xist* and reprogramming-resistant H3K9me3 (refs. [7,9]). However, it is possible that the H3K27me3-dependent imprinted genes combined with *Sfmbt2* miRNAs might have a cumulative effect on the development of SCNT embryos. Another possible explanation for the modest effect of *Sfmbt2* miRNA KO on the birth rate

is the persistence of functional abnormalities in miRNA KO SCNT placentas, although they recovered near-normal appearance. Gene expression analysis of KO SCNT placentas provided important insights into this effect, where only one GO term comprising "female pregnancy" was persistently enriched among the downregulated genes in E11.5 placentas in the wild type and KO SCNT (Supplementary Fig. 4). Importantly, this GO term includes the *Psg* family genes clustered on 7qA2/A3 (Supplementary Data 3). *Psg* genes encode pregnancy-specific glycoproteins that belong to the immunoglobulin superfamily, and they are secreted abundantly from trophoblastic cells into the maternal blood in hemochorial placentas to exert immunoregulatory and angiogenetic functions[35]. There are 10 human and 17 mouse *PSG* genes. In humans, low serum concentrations of PSGs are associated with fetal growth restrictions[43]. The

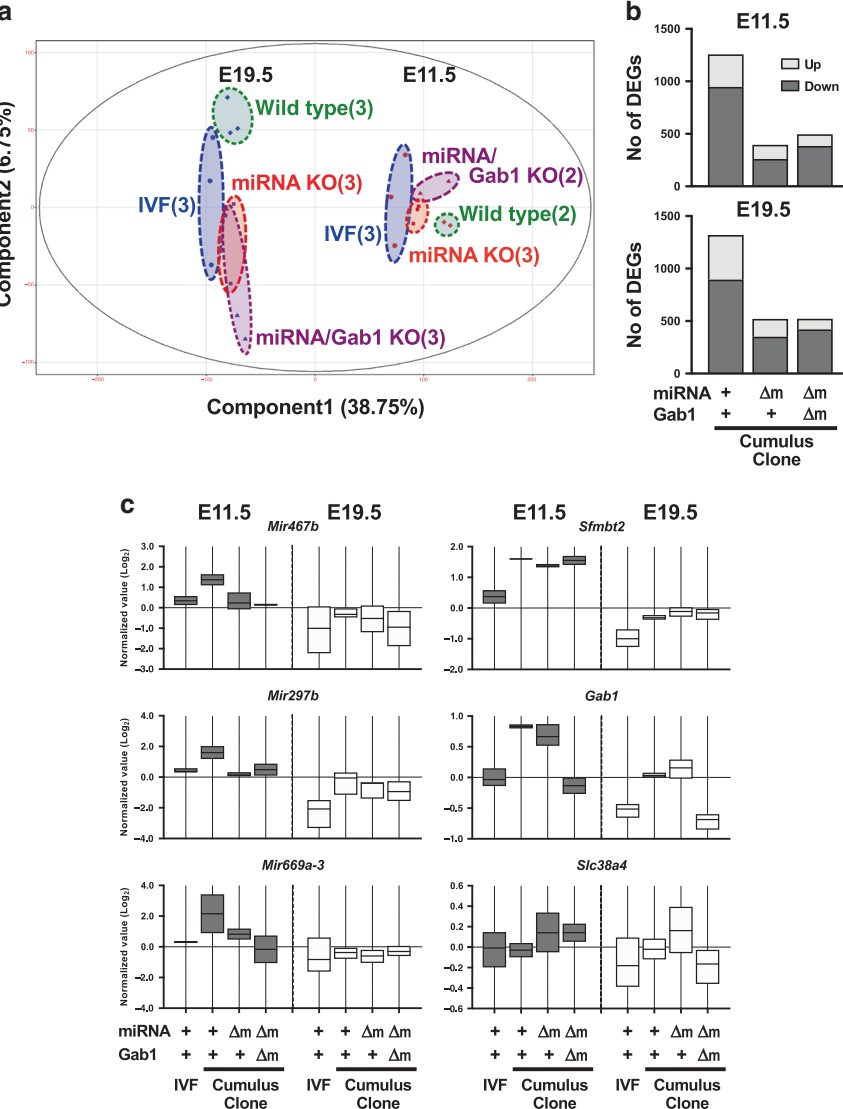

**Fig. 6 Transcriptome analysis of IVF and SCNT placentas from wild type and miRNA KO. a** Principal component analysis of IVF, SCNT wild type, miRNA KO, and miRNA/*Gab1* KO placentas at E11.5 and E19.5 (term). The numbers within brackets indicate the number of biological replicates. **b** Numbers of DEGs in SCNT (wild type and two types of KO) placentas vs IVF placentas. **c** Box plots showing the expression levels of *Sfmbt2* pri-miRNAs (*Mir467b*, *Mir297b*, and *Mir669a-3*) and placenta-specific imprinted genes (*Sfmbt2*, *Gab1*, and *Slc38a4*) in IVF and SCNT placentas. Upper, lower, and center lines indicate minimum, maximum, and mean values. The mean expression level in all samples is indicated as 0.0. +, wild type; Δm, maternal KO. Also see Supplementary Figs. 4 and 5, and Supplementary Data 2–6. Source data are provided as a Source data file.

consistent downregulation of *Psg* genes in mouse SCNT placentas might also have compromised fetal development through the disorganization of maternal–fetal interfaces.

Our findings have important implications for human and animal-assisted reproductive technologies (ARTs). In particular, they highlight the importance of the precise regulation of miRNA expression for normal placentation. In bovines, 278 out of the 377 miRNAs examined were downregulated in SCNT placentas, most of which were common to those detected in in vitro-produced placentas. Importantly, many of these miRNAs were colocalized on the same chromosomes as the clusters within or near imprinted regions[44], which is similar to the situation observed for *Sfmbt2* miRNAs and *Mirg* miRNAs. The primate-specific chromosome 19 miRNA cluster (C19MC) with imprinted paternal expression in the placenta may cause preeclampsia when aberrantly expressed by ARTs, such as in frozen-thawed blastocyst transfer[45]. It is known that ARTs may induce imprinting disorders in humans[46–48]. Many miRNAs exist as clusters within or near imprinted genes, so it is

likely that their expression is also affected by the dysregulation of host imprinted genes. Thus, it would be beneficial to examine how miRNAs are expressed in embryos and placentas generated by ARTs in humans and animals.

## Methods

**Animals.** Animals were provided with water and commercial laboratory mouse chow ad libitum, and were housed under controlled lighting conditions (daily light from 07:00 to 21:00). They were maintained under specific pathogen-free conditions. The care and use of animals in this study was performed according to the guidelines for the use and maintenance of experimental animals from the Japanese Ministry of Environment. All animal experiments included in this study were approved by the Institutional Animal Care and Use Committee of RIKEN Tsukuba Branch.

Eight- to 10-week-old C57BL/6 (B6) or (B6 × DBA/2) F1 (BDF1) female mice were used to collect oocytes or as nuclear donors. Eight- to 12-week-old ICR female mice were used as embryo transfer recipients. *Sfmbt2* miRNA KO and *Slc38a4* KO mouse lines were established using the CRISPR/Cas9 system, and maintained in the housing conditions described above[32,49]. *Sfmbt2* KO mice were generated using the CRISPR/Cas9 system in this study (see below). *Gab1* KO mice[18] were provided by RIKEN BRC through the National BioResource Project of MEXT/AMED, Japan

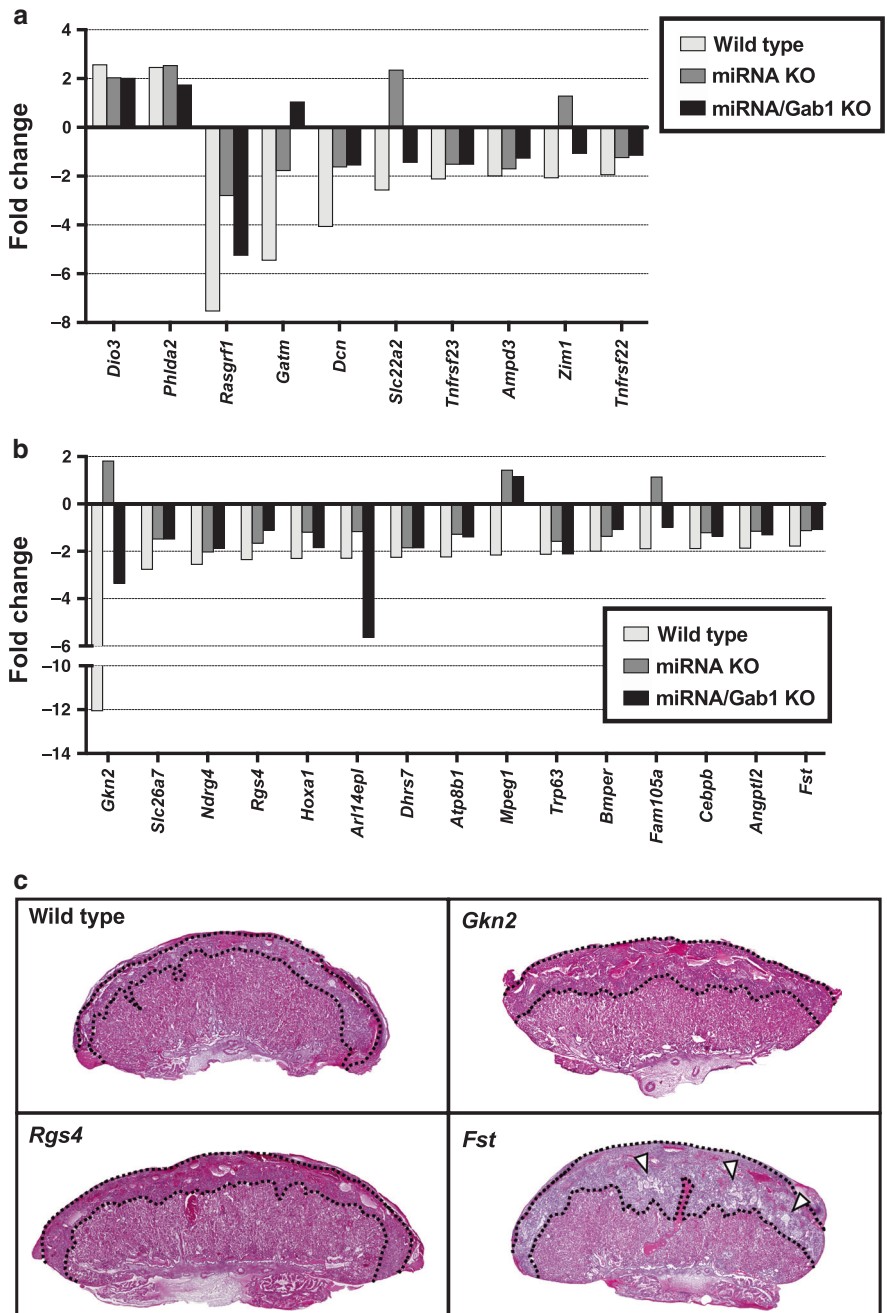

**Fig. 7 Changes of canonical imprinted genes and predicted target genes in SCNT placentas. a, b** Fold changes in the expression levels of canonical imprinted genes **a**, and the predicted target genes of *Sfmbt2* miRNAs **b** with different expression levels in IVF and SCNT placentas. The expression level in IVF placentas was set at 0. **c** Hematoxylin and eosin-stained tissue sections of E19.5 placentas from wild type and KO (*Gkn2*, *Rgs4*, and *Fst*) IVF. Scale bar, 1 mm. An increase in the number of glycogen cells was observed in the *Fst* KO placenta (arrowheads).

(RBRC00440). The *Gab1* KO allele was confirmed by PCR using the following primer sets: Gab1 geo 1443Lower: 5′-tcctccaccctgggggttcgtgtcctac-3′, Gab1 1356Lower: 5′-ggtaaagcaggtctaggtggctgacagtct-3′, and Gab1 1178Upper: 5′-gcgccttctttgcatcacctcatct-3′ (ref. [18]). *Gkn2* heterozygous KO mice were obtained by the embryo transfer of in vitro-fertilized embryos between wild type B6 oocytes and spermatozoa from *Gkn2* KO mice provided by KOMP Repository (C57BL/6N-Gkn2tm1(KOMP)Vlcg). Homozygous *Gkn2* KO mice were confirmed by PCR in the F1 offspring using the primer sets Gkn2 TUF: 5′-tgggtgattacctctgcttgttg-3′ and Gkn2 TUR: 5′-ttgcctcagtgtcttcaagtttac-3′ for the target gene; and KOMP LacINF: 5′-ggtaaactggctcggattaggg-3′ and KOMP LacINR: 5′-ttgactgtagcggctgatgttg-3′ for the reporter gene.

**IVF and nuclear transfer**. B6 or BDF1 female mice were induced to superovulate by injecting with 7.5 IU of pregnant mare serum gonadotropin (ZENOAQ) and 7.5

IU of human chorionic gonadotropin (hCG, Aska-Pharmaceutical) at an interval of 48 h.

Spermatozoa were collected from the epididymis of 12-week-old DBA/2 male mice and preactivated with human tubal fluid (HTF) medium for 1 h at 37 °C under 5% $CO_2$ in humidified air. Cumulus–oocyte complexes (COC) were collected from the oviducts of superovulated females and transferred to HTF. Preactivated spermatozoa were transferred into the oocyte culture medium at concentrations of 200–400 spermatozoa/μl. After culture for 6 h, pronuclear-stage embryos were transferred into potassium-enriched simplex optimized medium (KSOM) and cultured by embryo transfer.

Nuclear transfer was performed according to previous reports[50,51], with slight modifications. Briefly, at 15 h after injecting hCG, COC were collected from the oviducts and the cumulus cells were dispersed in KSOM containing 0.1% bovine testicular hyaluronidase. The collected cumulus cells were used as nuclear donors. The oocytes were enucleated in HEPES-buffered KSOM containing 7.5 μg/ml of

**Table 1 Developmental efficiency of SCNT embryos derived from donor cells with *Sfmbt2* miRNA KO or placenta-specific imprinted genes KO.**

| Genotype of donor cells | KO allele | N** | No. of cultured embryos | No. of cleaved embryos (%) | No. of embryos transferred (ET) | No. of recipient females | No. of implantations (%/ET) | No. of live pups born (%/ET) | No. of placentas without pups (%/ET) |
|---|---|---|---|---|---|---|---|---|---|
| Wild type | – | 3 | 167 | 152 (91.0) | 101 | 4 | 48 (47.5) | 3 (3.0) | 4 (4.0) |
| *Sfmbt2* KO | m or p | 6 | 382 | 364 (94.3) | 234 | 12 | 97 (41.5) | 6 (2.6) | 4 (1.7) |
| *Gab1* KO | m | 2 | 277 | 225 (81.2) | 203 | 9 | 75 (36.9) | 3 (1.5) | 7 (3.4) |
| *Sfmbt2* miRNA KO | m | 2 | 102 | 88 (86.3) | 75 | 3 | 27 (36.0) | 5 (6.7) | 2 (2.7) |
| *Sfmbt2* miRNA/*Gab1* KO | m/m | 2 | 187 | 164 (87.7) | 134 | 5 | 60 (44.8) | 9 (6.7) | 3 (2.2) |
| *Slc38a4* KO* | m | 2 | 84 | 79 (94.0) | 40 | 2 | 21 (52.5) | 3 (7.5) | 1 (2.5) |
| Wild type* | – | 3 | 201 | 194 (96.5) | 179 | 9 | 110 (61.5) | 15 (8.4) | 3 (1.7) |

*Embryos injected with *Kdm4d* mRNA instead of treatment with trichostatin A/latrunculin A. Data for the wild type are those reported by Matoba et al.[16].
**Number of experimental replicates.

cytochalasin B. The donor nuclei were injected into enucleated oocytes using a piezo-driven micromanipulator (Primetech). After culture in KSOM for 1 h, the injected oocytes were cultured for 8 h in $Ca^{2+}$-free KSOM containing 2.5 mM $SrCl_2$, 50 nM trichostatin A (Sigma-Aldrich), and 5 μM latrunculin A (Sigma-Aldrich).

**Embryo transfer and collection of placentas.** Fertilized or reconstructed embryos that reached the two-cell stage after culture for 24 h were transferred into the oviducts of pseudopregnant ICR female mice on day 0.5 (the day following sterile mating). On day 11.5, embryos were retrieved from the mother's uterus and placentas without deciduas were used in subsequent experiments. On day 19.5, the pregnant females underwent Caesarian section and placentas were collected. The live pups were nursed by lactating ICR females.

**Quantitative RT-PCR.** Reverse transcription and qRT-PCR were performed for miRNAs using a TaqMan miRNA reverse transcription kit (Thermo Fisher) for miRNA and Superscript IV (Thermo Fisher) for mRNA. For quantification of miRNA, QuantiTect Probe PCR kits (QIAGEN) with primers from TaqMan MicroRNA Assays (Thermo Fisher) were used and the quantified Ct values were normalized against that of the U6 snRNA, using the ΔΔCt method. Assay IDs were as follows: 001671 for miR467b-5p, 464896_mat for miR466b-3p, and 000452 for miR127, 000191 for miR300, and 001973 for U6 snRNA. Each experiment was performed in duplicate. For quantification of mRNA, QuantiTect SYBR Green PCR Kits (QIAGEN) were used and the quantified Ct values were normalized against that of β-actin, using the ΔΔCt method. All primer sequences used in this experiment are shown in Supplementary Data 8.

**Immunoblotting.** Protein lysates (20 μg) prepared from E19.5 placentas were separated by sodium dodecyl sulfate–10% polyacrylamide gel electrophoresis. The proteins were transferred to polyvinylidene difluoride membranes (GE Healthcare) electrophoretically using a Trans Blot-Turbo transfer system (Bio-Rad). Anti-CEBPB (ab32358, Abcam) and anti-actin (sc-1616, Santa Cruz Biotechnology) antibodies were diluted in phosphate-buffered saline (PBS) containing 0.2% Tween-20 (PBST) and 0.2% skim milk (1:1000, 1:5000, or 1:1000), followed by incubation at 4 °C overnight. The membranes were washed with PBST; incubated with horseradish peroxidase (HRP)-conjugated donkey anti-rabbit IgG (1:5000, AP182P, Millipore Corp) and HRP-conjugated donkey anti-goat IgG (1:5000, AP180P, Millipore Corp) at room temperature (RT) for 1 h; and washed with PBST. Signals were visualized using the ECL Prime Western Blotting detection reagent (GE Healthcare).

**Tissue sectioning.** Whole embryos (E11.5) or term placentas (E19.5) were fixed with Bouin's solution, and processed for paraffin sectioning. Sections with a thickness of 4 μm were stained with hematoxylin and eosin.

**RNA preparation and hybridization, and generation of the transcriptome library.** Total RNA was extracted from E11.5 and 19.5 placentas, using IVF and SCNT placentas with RNeasy Mini kits (QIAGEN). For microarray analysis, total RNA was labeled with an miRNA Complete Labeling and Hyb Kit (Agilent) and hybridized with SurePrint G3 Mouse miRNA 8 × 60 K (Agilent), according to the manufacturer's instructions. The transcriptome library was prepared using a SMARTer Stranded Total RNA Sample Prep Kit-HI Mammalian (Clontech), according to the manufacturer's instructions.

**miRNA microarray.** Scanning of microarray slides was performed with a DNA microarray scanner at a resolution of 5 μm (Agilent). Scanned image files were processed to obtain signal intensities using Feature Extraction software (Agilent). All raw data were loaded into Gene Spring GX software (Agilent) and normalized using the default settings. The microarray data were deposited in the Gene Expression Omnibus (GEO) under series accession number GSE129940.

**Transcriptome analysis.** Sequencing (100 base pair paired-end sequencing) was performed using the Illumina HiSeq 2500 platform (Illumina). All raw reads were loaded into Strand NGS (Agilent) and low-quality reads (QC < 20) were removed before further operations. Retained reads were aligned with the reference mouse genome (mm10), which was downloaded from the UCSC database using the default algorithm in Strand NGS after trimming the adapter sequences. Raw values were normalized using DE-seq and a baseline treatment was performed to set the mean value within each gene to zero. GO and chromosomal location analyses were performed using the DAVID Bioinformatics Resources (https://david.ncifcrf.gov).

**Target prediction.** Upregulated genes in E11.5 *Sfmbt2* miRNA KO placentas compared with the IVF placentas were extracted using the gene expression analysis method described in our previous study[32] (GSE82055). Genes with target sequences for the *Sfmbt2* miRNAs in their 3′-UTR were selected using TargetScan (http://www.targetscan.org/mmu_72/) and microRNA.org (http://www.microrna.org/microrna/home.do). The 102 common genes that met both criteria were

selected as the predicted target genes of *Sfmbt2* miRNAs. In the present study, the genes targeted in SCNT placentas were selected as the intersection of the genes downregulated in E11.5 wild type SCNT placentas and the 102 predicted target genes.

**In situ hybridization**. An E11.5 mouse placenta was fixed with G-Fix, embedded in paraffin, and sectioned at a thickness of 8 μm. In situ hybridization was performed using an ISH Reagent Kit according to the manufacturer's instructions. Tissue sections were deparaffinized with G-Nox (Genostaff) and rehydrated using an ethanol series and PBS. The sections were fixed with 10% formalin in PBS (NBF) for 15 min at RT and washed with PBS, treated with 4 μg/ml of proteinase K in PBS for 10 min at 37 °C and washed with PBS, refixed with 10% NBF for 15 min at RT and washed with PBS, and placed in 0.2 N HCl for 10 min at RT and washed with PBS. The sections were heat-treated in PBS for 5 min at 80 °C, cooled immediately in cold PBS, and placed in a Coplin jar containing 1× G-Wash (Genostaff). Hybridization was performed using probes at concentrations of 10 nM in G-Hybo or G-Hybo-L (Genostaff) for 16 h at 60 °C. After hybridization, the sections were washed three times with 10% formamide in 2× G-Wash for 30 min at 60 °C. The sections were then washed five times in 0.1% Tween-20 in TBS (TBST) at RT. After treatment with 1× G-Block (Genostaff) for 15 min at RT, the sections were incubated with anti-DIG AP conjugate (11093274910, Sigma-Aldrich) diluted to 1:2000 with G-Block in TBST for 1 h at RT. The sections were washed twice in TBST and incubated in 100 mM NaCl, 50 nM MgCl$_2$, 0.1% Tween-20, and 100 mM Tris-HCl at pH 9.5. Coloring reactions were performed using NBT/BCIP solution (Sigma-Aldrich) overnight, before washing with PBS and mounting using G-Mount (Genostaff). Probe IDs were 339111 for mmu-miR-669f-3p and 99004-15 for Scrambled-miR (Exiqon).

**Generation of CRISPR/Cas9 mice**. sgRNAs were designed using CRISPR design (the site was already closed) and produced using a GeneArt Precision gRNA Synthesis Kit (Thermo Fisher). Target sequences were as follows: agataaccggctcagcaaat and tctgcatggcacagcatcgt for *Sfmbt2* up- and downstream, respectively. sgRNA (50 ng/μl) and Alt-R S.p. Cas9 Nuclease (100 ng/μl, Integrated DNA Technologies) diluted with nuclease-free H$_2$O were microinjected into the cytoplasm of BDF1 × B6 or B6 × B6 in vitro-fertilized embryos. Injected embryos that reached the two-cell stage after culture for 24 h were transferred into the oviducts of pseudopregnant ICR female mice on day 0.5. On day 19.5, the pups were delivered by Caesarean section and the live pups were nursed by lactating ICR female mice. The sequences of the deleted genes of *Sfmbt2* (68 kb) were detected by PCR with the primer sets (Sfmbt2 F22813: 5′-ttgccagtttcgaagaagcg-3′, Sfmbt2 R91005: 5′-cctggactactccaacccca-3′) and deletions of the regions were confirmed by sequencing the PCR products. The founder mice obtained were crossed with B6 wild type mice for two or three generations and used in further experiments.

**Generation of triple-target CRISPR**. Triple-target CRISPR was performed according to a previous report[38], with slight modifications. sgRNAs of target genes were designed using CRISPR design and produced using a GeneArt Precision gRNA Synthesis Kit. sgRNA (150 ng/μl) and Alt-R S.p. Cas9 Nuclease (125 ng/μl) diluted with HEPES-KSOM were electroporated into BDF1 × B6 in vitro-fertilized embryos with a NEPA21 electroporator (NEPA gene). Treated embryos were cultured in KSOM for 18 h and embryos that reached the two-cell stage were transferred into the oviducts of pseudopregnant ICR female mice on day 0.5. On day 19.5, pups were delivered by Caesarean section, and the pups and placentas were collected. The CRISPR target sequences are listed in Supplementary Data 7.

**Statistical analysis**. DEGs were selected from the miRNA microarray data using Gene Spring GX. Statistically significant DEGs between IVF ($n = 4$) and cumulus-derived ($n = 3$) or Sertoli-derived ($n = 3$) cloned placentas were extracted with the moderated $t$-test, before applying the Benjamini and Hochberg false discovery rate procedure and requiring a change >1.3-fold. The Kruskal–Wallis test followed by Dunn's multiple comparison test were used to analyze the placental weights shown in Figs. 1a and 5a. Two-way ANOVA followed by Dunnett's multiple comparison test were applied to the data in Fig. 5b. The numbers of IVF, wild type, miRNA KO, and miRNA/*Gab1* KO placentas measured were seven, four, three, and four, respectively. The transcriptome analysis results shown in Figs. 6 and 7, and Supplementary Fig. 5 were obtained using Strand NGS. Genes with low expression levels and raw values < 20 in all samples were excluded from further analysis. Retained raw values were normalized with DE-seq and baseline treatment was performed using the default settings in the software. Genes with differences in expression >1.8-fold in the IVF and each of the SCNT placenta groups were designated as DEGs. The numbers of IVF, wild type, miRNA, and miRNA/*Gab1* placentas analyzed at E11.5 were three, two, three, and four, respectively. Three placentas were analyzed for each genotype at E19.5. The Kruskal–Wallis test followed by Dunn's multiple comparison test were applied to the data shown in Supplementary Fig. 3. *T*-tests were applied to the data in Supplementary Fig. 7a.

All error bars in the graphs represent the SEM and the horizontal bars indicate the mean values. In all statistical analyses, significant differences were accepted at $P < 0.05$.

**Reporting summary**. Further information on research design is available in the Nature Research Reporting Summary linked to this article.

## Data availability
miRNA microarray and RNA-seq data were deposited in NCBI GEO, under the accession code GSE129940. The source data underlying Figs. 1a, 3c, 5a, b, and 6c, and Supplementary Figs. 1, 2c, 3, 5, and 7a, b are provided as a Source data file. The relevant data that support the findings of this study are available from the corresponding authors upon reasonable request.

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

## Acknowledgements

This research was supported by KAKENHI (grant numbers 16H04687 (KI), 25112009, and 19H05758 (A.O.)), the Naito Foundation (K.I.), and Epigenome Manipulation Project of the all-RIKEN projects (A.O.). We thank T. Tomishima for her help with maintaining the mouse strains.

## Author contributions

K.I. and A.O. designed the research, and K.I. analyzed all the data. K.I., N.O., S.K., H.I., and S.M. produced the SCNT embryos. K.I. and S.M. designed the CRISPR/Cas9 sgRNA, and produced the KO strain for the *Sfmbt2* exon and *Slc38a4*, respectively. A.Ho. designed the CRISPR/Cas9 sgRNA for the *Sfmbt2* miRNA KO strain. K.Mi. analyzed the placental tissue sections. M.Hi. and M.Ha. analyzed the gene expression data from the TSC lines. A.Ha., N.W., Y.D., and K.Mo. maintained the mouse strains. K.I. and A.O. prepared the manuscript.

## Competing interests

The authors declare no competing interests.
