## [Peer Review File · Nature Communications]

Reviewers' comments:

Reviewer #1 (Remarks to the Author):

Review of Nature's Manuscript 213412_0, "Loss of H3K27me3 imprinting in cloned mouse placenta compromises normal placental development through upregulation of a large imprinted miRNA cluster", by Inoue et al.

The manuscript tackles a developmental abnormality that has accompanied SCNT since its beginning: placental dysfunction in clones. The manuscript provides robust evidence, gained with KO lines used for SCNT, and developmental/phenotypic/molecular studies, that the loss of H3K27me3 imprinting is the epigenetic error that compromises placenta function, and thus embryo development following SCNT. The experimental work and the ensuing results are carefully arranged and logically discussed. The outcome is a remarkable advancement in the knowledge of nuclear reprogramming through SCNT, and surely deserves to be published on Nature communications.

Minor points:

Page 3, Line 57: Somatic cell nuclear transfer (SCNT) is the only reproductive technique that can generate animals from single cells... not really, an oocyte is also required

Page 3, Line 63: the fact that the 129 strain skip the abnormal phenotype is intriguingly: is there any difference in the genes expression/imprinting of these genes addressed in this manuscript? If so, please add in the Discussion.

Page 6, Line 155-6: puzzling in here the differential expression of the Mirg miRNAs between the two cell donors, Sertoli and granulosa: are there any explanations the authors would like to provide?

Page 12, Discussion: Sfbmt2 miRNAs have genomic wide impact: cell cycle control, cell proliferation, pluripotency. Do the Authors notice any phenotypic effect on the cloned offspring?

Reviewer #2 (Remarks to the Author):

In this paper, the authors demonstrate that the loss of H3K27me3 imprinting, occurring during somatic cell nuclear transfer (SCNT), leads to altered expression of a family of placenta-specific miRNAs known to regulate placental development.

The authors build upon their previous work and designed a new set of genetic studies involving a number of genetic knockouts and transgenic animals to carefully analyze the contribution of various genetic entities to the development of placenta in the context of SCNT.

In particular, they provide strong evidence that the protein-coding genes that are dysregulated in SCNT placentas due loss of imprinting do not contribute much to the phenotype observed. Instead, a set of miRNAs that are also dysregulated seem to be the leading cause of the defects.

Altogether, the manuscript is well written, and the quality of the work is excellent. The authors present a lot of interesting data and the conclusions will likely interest many people in the field. I have only a few minor comments.

1. Ablation of the Sfmbt2 miRNA cluster apparently did not affect the expression of the Sfmbt2 mRNA. However, it would be prudent to make sure that the protein levels of Sfmbt2 are also not affected.

2. The in situ hybridization pictures in fig. 4 of the miRNA are suboptimal. There seem to be a lot of punctate dots in panel B, and some of them are outside the region of the giant trophoblast cells. It would be helpful to show other regions of the placenta containing giant cells and see a better demarcation between the labeled and non-labeled nuclei.

In the result section, lines 171- 172, the authors wrote: “Positive staining of nuclei was identified in secondary trophoblast giant cells (Fig. 4b), which was consistent with the previously reported localization of the SFMBT2 protein”. However, the fact that the SFMBT2 protein ends up in the nucleus is no guarantee that the intragenic miRNAs will exhibit the same subcellular localization. The processes involved in the trafficking of the two species are usually independent.

Also, in fig. 4, panel C, the staining is of low intensity and does not seem to be different between the ST layer and the LB layer, which contradicts the claim that Sfmbt2 miRNAs are expressed predominantly in the ST layer. Some clarification is needed here.

3. The claim that Gab1 might act synergistically with the Sfmbt2 miRNAs is not entirely convincing. The effect described, on placenta weight, is rather subtle. Did the difference between miRNA KO placentas and miRNA/Gab1 double KO reach statistical significance?

Besides, in fig.6, panel A, the principal component analysis shows that the transcription profile from miRNA KO samples is more similar to the IVF's than is the miRNA/Gab1 double KO. All this suggests that there might be some overlap in the regulatory networks, but “synergy” seems to be an overstatement.

Reviewer #3 (Remarks to the Author):

This study addresses a defined question of the molecular defects that contribute to enlarged placenta following somatic cell nuclear transfer (SCNT) in mice. The authors demonstrate that a major contributing factor is the anomalous expression of miRNAs encoded within the placenta-specific imprinted Sfmbt2 locus, one of the genes recently determined to be regulated by ‘non-canonical’ imprinting, i.e., dependent on repressive H3K27me3 in oocytes rather than DNA methylation. The manuscript represents a substantial amount of work, including genetic analysis to test whether the dosage (loss of imprinting) of selected genes contributes to placental overgrowth and poor outcomes of SCNT, as well as CRISPR/Cas9 ablation of target genes of the Sfmbt2-encoded miRNAs. The present study builds upon previous work from the group on understanding the molecular defects in SCNT-derived placenta, as well as work from the Yi Zhang group that identified non-canonical imprinting as a parallel mechanism of imprinting that becomes restricted to placental lineages. It helps provide insight into the significance of this new form of imprinting in placental development and could benefit future applications of SCNT, although the relevance of the findings to other species may need to await a better understanding of how conserved these genes and their imprinting are in other species.

The authors identify significant upregulation of miRNAs encoded within *Sfmbt2* in SCNT placenta. They are able to exploit some existing genetic knock-out models (as well as generating a new one) to show that gene dosage of three protein-coding imprinted (non-canonical) genes does not contribute to placenta defects, but that dosage of the *Sfmbt2*-encoded miRNAs does (using a knock-out allele they recently generated). They identify predicted targets of the *Sfmbt2* miRNAs and by RNA-seq identify that some of these are down-regulated in correlation with up-regulation of the miRNAs. Finally, they undertake CRISPR/Cas9 in vivo knock-outs of selected miRNA targets, reporting that ablation of several of these genes individually led to altered placental phenotypes.

As said, the study represents a substantial amount of work, much of which is elegant. However, there are some major concerns mostly relating to validation of some of the knock-out models used.

Here are the major points:

1. Figure 1 reports that the overgrowth characteristic of SCNT placenta is not rescued when SCNT is performed with cells carrying maternal deletions in the protein coding genes *Sfmbt2*, *Gab1* or *Slc38a4*. The logic is that cloning from somatic cells (cumulus or Sertoli in this case) circumvents the normal mechanism of imprinting of these genes (by H3K27me3 from the oocyte), such that these genes lack imprinting and are two-fold up-regulated in SCNT placenta. Therefore, by cloning from cells in which one of the expressed alleles is deleted, normal gene expression dosage should be restored. As far as I can see, however, this assumption is not tested in the current study, and the authors should provide expression data to demonstrate that normal expression levels are rescued, otherwise it is a negative result with an ambiguous interpretation. Note that later on in the manuscript the authors report that *Slc38a4* expression is not actually upregulated in SCNT placenta (Fig. 6c).
2. Figure 4: the in situ result seems to be so weak that it is difficult to conclude that there is a specific expression signal for mi467b-5p. I am sure this is a technically demanding experiment, but it is not clear that this equivocal result adds anything to the manuscript.
3. Figure 7 and Supplementary Figure 5: these show gross placental phenotypes of CRISPR-generated knock-outs of predicted target genes of the *Sfmbt2*-miRNAs. They appear to be 'transient' gene ablations in which the genes of interest have been targeted in electroporated zygotes/embryos and placenta evaluated close to term. However, the authors present no validation that the targeted genes have actually been ablated, whether this is a fully-penetrant ablation or mosaic, how many ablated embryos/placenta were recovered and how many were assessed for their placental histology. Without this sort of basic validation, it is impossible to interpret the results shown.
4. Supplementary Tables 1–6: show the differentially expressed miRNAs or mRNAs from the various experiments. The tables report a summary statistic only: fold-change (FC) for each of the differentially expressed miRNAs/mRNAs identified. To my mind, these supplementary tables should include the expression values for each of the replicates, as well as the mean and then FC as summary statistic, and

adjusted p-values. I think this is essential additional information to be able to assess the consistency of the effects. For clarification, are the fold changes values log-transformed?

Minor points:

1. Figure 3a,b and Supplementary Figure 1a,b: these plots need clearly labelled axes. Also, it would be preferable if these scatterplots were squares with the x and y axes the same lengths rather than rectangles.

2. Figure 5a: "Therefore, although Gab1 alone had little effect, it might have synergistically affected the SCNT placentas in the presence of the Sfbmt2 miRNAs". A statistical test between the miRNA and miRNA/Gab1 knock-out would seem to be obvious here and is missing.

3. Figure 6c and Supplementary Figure 4: "Other placenta-specific imprinted genes (Smoc1 and Phf17) were also upregulated in E11.5 SCNT placentas (Supplementary Fig. 4), whereas Slc38a4 was not (Fig. 6c). Thus, deletion of the maternal allele from the Sfbmt2 miRNA cluster corrected the expression levels of miRNAs in SCNT placentas, but it did not affect the expression levels of the placenta-specific imprinted genes, including the host Sfbmt2 gene." This seems to be an oversimplification of the actual results presented; for example, from Suppl. Fig. 4 it seems that the expression level of the Smoc1 genes is restored to normal (IVF) levels in SCNT placenta with the miRNA/Gab1 knock-outs at e19.5, or for Phf17 at e11.5. It is not entirely clear why these gene expression 'corrections' might occur, but it could also indicate some level of inconsistency in the RNA-seq data.

4. Table 1: "The birth rates of the clones from miRNA KO (6.7%) and miRNA/Gab1 double KO (6.7%) donor cells were more than double those of the clones from WT donors (3.0%).....(Table 1)." It is not clear where these figures are in Table 1, which does not appear to give live birth outcomes. Also, in this Table, what does 'N' refer to, what does 'Fetus (%)' refer to, and what does 'Placenta only (%)' refer to?

In general in the manuscript there is little information about the number of biological replicates for many of the measures.

Response to the reviewers

We wish to thank the reviewers for their helpful suggestions and to express our appreciation of their understanding of the significance of our study. Our replies to the reviewers' comments are given below.

Reviewer #1

The manuscript tackles a developmental abnormality that has accompanied SCNT since its beginning: placental dysfunction in clones. The manuscript provides robust evidence, gained with KO lines used for SCNT, and developmental/phenotypic/molecular studies, that the loss of H3K27me3 imprinting is the epigenetic error that compromises placenta function, and thus embryo development following SCNT. The experimental work and the ensuing results are carefully arranged and logically discussed. The outcome is a remarkable advancement in the knowledge of nuclear reprogramming through SCNT, and surely deserves to be published on Nature communications.

We thank Reviewer #1 for the careful review and very helpful suggestions. We are pleased to know that the referee understood the significance of our study and evaluated it highly. All the suggestion and comments have been considered and incorporated into the revised manuscript. For the changes made from the previous version, please see a marked-up copy file.

Minor points:

Page 3, Line 57: Somatic cell nuclear transfer (SCNT) is the only reproductive technique that can generate animals from single cells.... not really, an oocyte is also required

We have added the word "oocyte" in the following sentence. "Somatic cell nuclear transfer (SCNT) is the only reproductive technique that can generate animals from oocytes and single cells (Page 3, Line 58)."

Page 3, Line 63: the fact that the 129 strain skip the abnormal phenotype is intriguingly: is there any difference in the genes expression/imprinting of these genes addressed in this manuscript? If so, please add in the Discussion.

We thank the reviewer for pointing out the text that describes strain 129. However, currently, we do not have data that explain the differences in the imprinting or gene expression patterns observed between strain 129 and other strains in relation to the placental phenotypes. Thus, we have just added “for unknown reasons” at the end of the sentence (Page 3, line 64).

Page 6, Line 155-6: puzzling in here the differential expression of the Mirg miRNAs between the two cell donors, Sertoli and granulosa: are there any explanations the authors would like to provide?

We do not have any data that can answer this question convincingly. However, in our preliminary study, the expression of *Meg3* in the *Dlk1–Dio3* imprinted domain exhibited the same tendency detected for the *Mirg* miRNA in SCNT placentas (please refer to the figure below, for reviewers only). Thus, although common mechanisms may be shared by them, the gene expression machineries at the *Dlk1–Dio3* domain are complicated by the involvement of the 3D genomic structure and the presence of secondary DMRs (e.g., <https://www.biorxiv.org/content/10.1101/633065v1>). Therefore, we have added “probably because of some regulatory mechanisms specific to the *Dlk1–Dio3* imprinted domain.” at the end of the sentence (Page 6, Line 162).

Page 12, Discussion: Sfmt2 miRNAs have genomic wide impact: cell cycle control,

cell proliferation, pluripotency. Do the Authors notice any phenotypic effect on the cloned offspring?

According to our observations, the cloned offspring were indistinguishable from IVF-derived pups in appearance and grew into adults without any abnormal phenotypes. We have added birth-weight data and a description of the normal phenotypes of adults, as follows.

“The average birth weight of miRNA KO pups was not different from that of IVF-derived or wild-type cloned pups (1.54 ± 0.04 g, 1.73 ± 0.10 g, and 1.63 ± 0.08 g in IVF-derived, wild-type, and miRNA KO pups, respectively). These miRNA KO cloned pups were normal in appearance and grew into normal fertile adults.” (Page 7, Line 193).

Reviewer #2

In this paper, the authors demonstrate that the loss of H3K27me3 imprinting, occurring during somatic cell nuclear transfer (SCNT), leads to altered expression of a family of placenta-specific miRNAs known to regulate placental development. The authors build upon their previous work and designed a new set of genetic studies involving a number of genetic knockouts and transgenic animals to carefully analyze the contribution of various genetic entities to the development of placenta in the context of SCNT. In particular, they provide strong evidence that the protein-coding genes that are dysregulated in SCNT placentas due loss of imprinting do not contribute much to the phenotype observed. Instead, a set of miRNAs that are also dysregulated seem to be the leading cause of the defects. Altogether, the manuscript is well written, and the quality of the work is excellent. The authors present a lot of interesting data and the conclusions will likely interest many people in the field. I have only a few minor comments.

We thank Reviewer #2 for the careful review and very helpful suggestions, especially for the comments on the result of *in situ* hybridization. We are pleased to know that the referee understood the significance of our study and evaluated it highly. All the suggestion and comments have been considered and incorporated into the revised manuscript. For the changes made from the previous version, please see a marked-up copy file.

1. Ablation of the *Sfmbt2* miRNA cluster apparently did not affect the expression of the *Sfmbt2* mRNA. However, it would be prudent to make sure that the protein levels of *Sfmbt2* are also not affected.

We appreciate the reviewer's suggestion. We purchased several commercially available mouse anti-SFMBT2 antibodies. However, these reagents were unreliable in terms of specificity. Thus, we generated *Sfmbt2* knock-in mouse lines with a Halo-tag and mated the resultant *Sfmbt2* Halo-tag males with *Sfmbt2* miRNA heterozygous KO females. Using the hybrid placentas obtained in this way, we compared the SFMBT2 protein levels in placentas with or without miRNA KO; no significant difference was detected between them. The data are shown in Supplementary Figure 4 and a sentence has been added to the main text, accordingly: "We confirmed that the level of the SFMBT2 protein in placentas was not affected by maternal *Sfmbt2* miRNA KO (henceforth, referred to as miRNA KO) using an SFMBT2 Halo-tag knock-in mouse line generated in-house (Supplementary Fig. 4)." (Page 7, Line 187).

2. The in situ hybridization pictures in fig. 4 of the miRNA are suboptimal. There seem to be a lot of punctate dots in panel B, and some of them are outside the region of the giant trophoblast cells. It would be helpful to show other regions of the placenta containing giant cells and see a better demarcation between the labeled and non-labeled nuclei.

We appreciate the reviewer's suggestion. We reinvestigated the localization of this miRNA using a new probe for *miR669f-3p*, which is one of the most highly expressed *Sfmbt2* miRNAs. The result of this experiment supported the finding described in the previous version of the manuscript: the miRNA was expressed in the secondary giant cells and trophoblasts in the spongiotrophoblast layer. We noted that it was also expressed in immature trophoblasts in the labyrinthine layer. New panels have been inserted in Figure 4 and the text has been modified accordingly. In contrast with the original figure, the contour of the nuclei of miRNA-expressing giant cells can be easily identified in the new figure.

In the result section, lines 171- 172, the authors wrote: “Positive staining of nuclei was identified in secondary trophoblast giant cells (Fig. 4b), which was consistent with the previously reported localization of the SFMBT2 protein”. However, the fact that the SFMBT2 protein ends up in the nucleus is no guarantee that the intragenic miRNAs will exhibit the same subcellular localization. The processes involved in the trafficking of the two species are usually independent.

We agree with the reviewer. According to his/her suggestion, we have deleted the following phrase from the main text: “which was consistent with the previously reported localization of the SFMBT2 protein¹⁷” (Page 7, Line 178 in the original version).

Also, in fig. 4, panel C, the staining is of low intensity and does not seem to be different between the ST layer and the LB layer, which contradicts the claim that *Sfmbt2* miRNAs are expressed predominantly in the ST layer. Some clarification is needed here.

We appreciate the reviewer's suggestion. As explained above, we used a new probe for an alternative highly expressed miRNA and obtained clearer images of *in situ* hybridization. Using this probe, both LB and ST layer cells were stained indistinguishably. Thus, we have

corrected the corresponding sentence (Page 7, Line 179).

3. The claim that *Gab1* might act synergistically with the *Sfmbt2* miRNAs is not entirely convincing. The effect described, on placenta weight, is rather subtle. Did the difference between miRNA KO placentas and miRNA/*Gab1* double KO reach statistical significance?

Besides, in fig.6, panel A, the principal component analysis shows that the transcription profile from miRNA KO samples is more similar to the IVF's than is the miRNA/*Gab1* double KO. All this suggests that there might some overlap in the regulatory networks, but “synergy” seems to be an overstatement.

Regarding these two questions, we performed a statistical analysis between these two groups and obtained a *P* value of 0.048 (*t*-test, nonparametric), indicating that the difference was significant (if significance is set at $P < 0.05$). However, the difference was only marginally significant, and we agree with the reviewer that the word “synergy” seems to be an overstatement. Thus, in the revised version, we have deleted the sentence “Therefore, although *Gab1* alone had little effect, it might have synergistically affected the SCNT placentas in the presence of the *Sfmbt2* miRNAs (Fig. 5a).” The *P* value calculated between miRNA and miRNA/*Gab1* KO has been added to the main text, as follows. “Intriguingly, the placental weight was decreased further under these conditions (0.17 ± 0.01 g, $P = 0.981$ vs. IVF, $P = 0.048$ vs. miRNA KO).” (Page 7, Line 198).

Reviewer #3 (Remarks to the Author):

This study addresses a defined question of the molecular defects that contribute to enlarged placenta following somatic cell nuclear transfer (SCNT) in mice. The authors demonstrate that a major contributing factor is the anomalous expression of miRNAs encoded within the placenta-specific imprinted Sfbmt2 locus, one of the genes recently determined to be regulated by ‘non-canonical’ imprinting, i.e., dependent on repressive H3K27me3 in oocytes rather than DNA methylation. The manuscript represents a substantial amount of work, including genetic analysis to test whether the dosage (loss of imprinting) of selected genes contributes to placental overgrowth and poor outcomes of SCNT, as well as CRISPR/Cas9 ablation of target genes of the Sfbmt2-encoded miRNAs. The present study builds upon previous work from the group on understanding the molecular defects in SCNT-derived placenta, as well as work from the Yi Zhang group that identified non-canonical imprinting as a parallel mechanism of imprinting that becomes restricted to placental lineages. It helps provide insight into the significance of this new form of imprinting in placental development and could benefit future applications of SCNT, although the relevance of the findings to other species may need to await a better understanding of how conserved these genes and their imprinting are in other species.

The authors identify significant upregulation of miRNAs encoded within Sfbmt2 in SCNT placenta. They are able to exploit some existing genetic knock-out models (as well as generating a new one) to show that gene dosage of three protein-coding imprinted (non-canonical) genes does not contribute to placenta defects, but that dosage of the Sfbmt2-encoded miRNAs does (using a knock-out allele they recently generated). They identify predicted targets of the Sfbmt2 miRNAs and by RNA-seq identify that some of these are down-regulated in correlation with up-regulation of the miRNAs. Finally, they undertake CRISPR/Cas9 in vivo knock-outs of selected miRNA targets, reporting that ablation of several of these genes individually led to altered placental phenotypes.

As said, the study represents a substantial amount of work, much of which is elegant. However, there are some major concerns mostly relating to validation of some of the knock-out models used.

We thank Reviewer #3 for careful review and very helpful comments. We are pleased to know that the reviewer understood the significance of our study and evaluated it highly. All the suggestions/comments have been considered and incorporated in the revised version by performing necessary experiments including validation of the knock-out models used. For the changes made from the previous version, please see a marked-up copy file.

Here are the major points:

1. Figure 1 reports that the overgrowth characteristic of SCNT placenta is not rescued when SCNT is performed with cells carrying maternal deletions in the protein coding genes *Sfmbt2*, *Gab1* or *Slc38a4*. The logic is that cloning from somatic cells (cumulus or Sertoli in this case) circumvents the normal mechanism of imprinting of these genes (by H3K27me3 from the oocyte), such that these genes lack imprinting and are two-fold up-regulated in SCNT placenta. Therefore, by cloning from cells in which one of the expressed alleles is deleted, normal gene expression dosage should be restored. As far as I can see, however, this assumption is not tested in the current study, and the authors should provide expression data to demonstrate that normal expression levels are rescued, otherwise it is a negative result with an ambiguous interpretation. Note that later on in the manuscript the authors report that *Slc38a4* expression is not actually upregulated in SCNT placenta (Fig. 6c).

We investigated the expression levels of these three genes by quantitative RT-PCR using IVF and cloned placentas at E19.5. The results of this experiment are shown in Supplementary Figure 1 and in the revised text (Page 4, Line 120). These three genes were overexpressed in cloned placentas and were rescued by the respective maternal deletion. As pointed out by the reviewer, *Slc38a4* was not upregulated in Figure 6c. We assume that this was caused by the great variation in the IVF-derived placentas. Our previous study demonstrated that *Slc38a4* was overexpressed in E13.5 cloned placentas (Okoe et al.). These explanations have been added to the text (Page 9, Line 250).

2. Figure 4: the in situ result seems to be so weak that it is difficult to conclude that there is a specific expression signal for mi467b-5p. I am sure this is a technically demanding experiment, but it is not clear that this equivocal result adds anything to the manuscript.

According to the reviewer's suggestion, we designed a new probe for an alternative highly expressed miRNA (*miR-669f-3p*) to obtain better *in situ* hybridization images. As a result, we obtained clearer images of the localization of the expressed miRNA. We have replaced Figure 4 with new figures and have modified the main text accordingly (from Page 9, Line 175).

3. Figure 7 and Supplementary Figure 5: these show gross placental phenotypes of

CRISPR-generated knock-outs of predicted target genes of the Sfbmt2-miRNAs. They appear to be ‘transient’ gene ablations in which the genes of interest have been targeted in electroporated zygotes/embryos and placenta evaluated close to term. However, the authors present no validation that the targeted genes have actually been ablated, whether this is a fully-penetrant ablation or mosaic, how many ablated embryos/placenta were recovered and how many were assessed for their placental histology. Without this sort of basic validation, it is impossible to interpret the results shown.

According to the reviewer’s suggestion, we examined whether the targeted genes were ablated by quantitative RT–PCR and Western blotting using CRISPR KO placentas. Using quantitative RT–PCR, we confirmed that the expression of all (nine) target genes was significantly decreased in KO placentas. Unfortunately, we were unable to find reliable commercially available antibodies in terms of specificity, to be used in Western blotting, with the exception of CEBPB. We confirmed the nearly complete ablation of the CEBPB protein. In the revised version, we have presented these results in Supplementary Figure 8 and have added relevant sentences to the main text (Page 10, Line 288). The number of placentas analyzed are shown in Supplementary Table 7. We did not analyze mosaicism in F0 founder mice because of technical difficulties. However, Western blotting for the CEBPB protein and the phenotypes (lethality) of *Fst* and *Bmper* KO neonates indicate that the contribution of the intact allele might have been negligible, at least for KOs of these three genes.

4. Supplementary Tables 1–6: show the differentially expressed miRNAs or mRNAs from the various experiments. The tables report a summary statistic only: fold-change (FC) for each of the differentially expressed miRNAs/mRNAs identified. To my mind, these supplementary tables should include the expression values for each of the replicates, as well as the mean and then FC as summary statistic, and adjusted p-values. I think this is essential additional information to be able to assess the consistency of the effects. For clarification, are the fold changes values log-transformed?

We appreciate the reviewer’s suggestion. We have modified Tables 1–6 by including these values. Fold-change values were not log-transformed.

Minor points:

1. Figure 3a,b and Supplementary Figure 1a,b: these plots need clearly labelled axes. Also, it would be preferable if this scatterplots were squares with the x and y axes the same lengths rather than rectangles.

As suggested by the reviewer, we have clarified the labeling of the axes and corrected these scatter plot figures to squares. (Figure 3a and b, Supplementary Figure 2a and b)

2. Figure 5a: “Therefore, although *Gab1* alone had little effect, it might have synergistically affected the SCNT placentas in the presence of the *Sfmbt2* miRNAs”. A statistical test between the miRNA and miRNA/*Gab1* knock-out would seem to be obvious here and is missing.

We performed a statistical analysis between these two groups and obtained a *P* value of 0.048 (*t*-test, nonparametric), indicating that the difference was significant (if significance was set at $P < 0.05$). However, the difference was marginal, and we agree with Reviewer 2 (Comment #3) that “synergy” seems to be an overstatement. Thus, in the revised version, we have deleted the sentence “Therefore, although *Gab1* alone had little effect, it might have synergistically affected the SCNT placentas in the presence of the *Sfmbt2* miRNAs (Fig. 5a).” from the main text. The *P* value obtained for the comparison between miRNA and miRNA/*Gab1* KO has been added to the main text, as follows. “Intriguingly, the placental weight was decreased further under these conditions (0.17 ± 0.01 g, $P = 0.981$ vs. IVF, $P = 0.048$ vs. miRNA KO).” (Page 7, Line 200).

3. Figure 6c and Supplementary Figure 4: “Other placenta-specific imprinted genes (*Smoc1* and *Phf17*) were also upregulated in E11.5 SCNT placentas (Supplementary Fig. 4), whereas *Slc38a4* was not (Fig. 6c). Thus, deletion of the maternal allele from the *Sfmbt2* miRNA cluster corrected the expression levels of miRNAs in SCNT placentas, but it did not affect the expression levels of the placenta-specific imprinted genes, including the host *Sfmbt2* gene.” This seems to be an oversimplification of the actual results presented; for example, from Suppl. Fig. 4 it seems that the expression level of the *Smoc1* genes is restored to normal (IVF) levels in SCNT placenta with the miRNA/*Gab1* knock-outs at e19.5, or for *Phf17* at e11.5. It is not entirely clear why these gene expression ‘corrections’ might occur, but it could also indicate some level of inconsistency in the RNA-seq data.

We agree with the reviewer that this inconsistency may have been caused by some level of

inconsistency in the RNA-seq data. Furthermore, the corrected expression of *Smoc1* in E19.5 placentas might be a result of improved placental morphology, even though the exact localization of *Smoc1* has not been identified. Thus, we have modified the original sentence as follows. “Other placenta-specific imprinted genes, *Smoc1* and *Phf17*¹⁵, were also upregulated in E11.5 SCNT placentas (Supplementary Fig. 6), whereas the expression levels of *Smoc1* in E19.5 and *Phf17* in E11.5 seemed to be corrected by miRNA KO and miRNA/*Gab1* KO placentas. Although the reasons for their corrected expression patterns are unknown, normalized placental histology (especially at E19.5) or corrected physiological conditions may have been at play.” (Page 9, Line 245).

4. Table 1: “The birth rates of the clones from miRNA KO (6.7%) and miRNA/*Gab1* double KO (6.7%) donor cells were more than double those of the clones from WT donors (3.0%).....(Table 1).” It is not clear where these figures are in Table 1, which does not appear to give live birth outcomes. Also, in this Table, what does ‘N’ refer to, what does ‘Fetus (%)’ refer to, and what does ‘Placenta only (%)’ refer to?

We apologize for the poor explanations provided for this table. As suggested by the reviewer, we corrected the terms in the table and added necessary explanations in the footnote.

- N, number of experimental replicates (added to the footnote)
- “Fetus” was changed to “No. of live pups born” (changed terms).
- “Placenta” was changed to “No. of placentas without pups” (changed terms).

In general in the manuscript there is little information about the number of biological replicates for many of the measures.

The biological replicates are indicated in the Methods, and the Reporting Summary; however, we have appropriately added biological replicates to the figures, for the readers’ convenience.

REVIEWERS' COMMENTS:

Reviewer #1 (Remarks to the Author):

The Authors have correctly addressed my comments; therefore, I recommend its publication on Nature Communications

Reviewer #3 (Remarks to the Author):

I think the authors have done a good job in addressing as far as practicable all the issues raised by myself and the other two reviewers.

The only issue that I do not think they have been able to address was that raised by Reviewer 2 point 1, regarding whether the miRNA-specific knock-out of the *Sfmbt2* locus had a detrimental effect on SFMBT2 protein levels. In the absence of reliable commercial antibodies, the authors sought to test this by generating a Halo-tagged *Sfmbt2* allele, and show that when crossed with the *Sfmbt2*-miRNA deletion allele that SFMBT2 protein levels are not altered. However, this tagged allele represents protein expression from the wild-type paternal allele, not the maternal miRNA-deletion allele, which is the relevant one to test, i.e., that the deletion altered processing or stability of the *Sfmbt2* transcript. The experiment might test whether the presence of excess copies of the miRNAs themselves altered the wild-type *Sfmbt2* transcript, but I don't think that was the question the reviewer was asking. Therefore, I don't believe the experiment was very meaningful and could be omitted from the manuscript. But I do accept that authors' comment that there are no suitable commercially available SFMBT2 antibodies that would allow them to test the question posed by the reviewer.

Gavin Kelsey

Response to the reviewers

We are grateful to all of the reviewers for their careful reading of, and supportive comments for our revised manuscript. Our replies to the reviewers' comments are indicated below.

Reviewer #1 (Remarks to the Author):

The Authors have correctly addressed my comments; therefore, I recommend its publication on Nature Communications

We appreciate the supportive comment of Reviewer #1.

Reviewer #3 (Remarks to the Author):

I think the authors have done a good job in addressing as far as practicable all the issues raised by myself and the other two reviewers.

The only issue that I do not think they have been able to address was that raised by Reviewer 2 point 1, regarding whether the miRNA-specific knock-out of the Sfmbt2 locus had a detrimental effect on SFBMT2 protein levels. In the absence of reliable commercial antibodies, the authors sought to test this by generating a Halo-tagged Sfmbt2 allele, and show that when crossed with the Sfmbt2-miRNA deletion allele that SFBMT2 protein levels are not altered. However, this tagged allele represents protein expression from the wild-type paternal allele, not the maternal miRNA-deletion allele, which is the relevant one to test, i.e., that the deletion altered processing or stability of the Sfmbt2 transcript. The experiment might test whether the presence of excess copies of the miRNAs themselves altered the wild-type Sfmbt2 transcript, but I don't think that was the question the reviewer was asking. Therefore, I don't believe the experiment was very meaningful and could be omitted from the manuscript. But I do accept that authors' comment that there are no suitable commercially available SFBMT2 antibodies that would allow them to test the question posed by the reviewer.

We are truly grateful to Reviewer #3 for carefully reading the whole manuscript and for providing constructive suggestions. According to this reviewer's recommendation, we omitted Supplementary Figure 4 and the relevant sentences (below) from the main text.

“We confirmed that the level of the SFMBT2 protein in placentas was not affected by maternal *Sfmbt2* miRNA KO (henceforth referred to as miRNA KO) using an SFMBT2 Halo-tag knock-in mouse line generated in-house (Supplementary Figure 4).”